# LEARNING TO MODULATE PRE-TRAINED MODELS IN RL

**Thomas Schmied**[*1]**, Markus Hofmarcher**[1,2]**, Fabian Paischer**[1]**, Razvan Pascanu**[3]**, Sepp Hochreiter**[1,4]

[1] ELLIS Unit Linz and LIT AI Lab, Institute for Machine Learning, Johannes Kepler University
[2] JKU LIT SAL eSPML Lab, Institute for Machine Learning, Johannes Kepler University
[3] DeepMind, London, UK
[4] Institute of Advanced Research in Artificial Intelligence (IARAI), Vienna, Austria

## ABSTRACT

Reinforcement Learning (RL) has experienced great success in complex games and simulations. However, RL agents are often highly specialized for a particular task, and it is difficult to adapt a trained agent to a new task. In supervised learning, an established paradigm is multi-task pre-training followed by fine-tuning. A similar trend is emerging in RL, where agents are pre-trained on data collections that comprise a multitude of tasks. Despite these developments, it remains an open challenge how to adapt such pre-trained agents to novel tasks while retaining performance on the pre-training tasks. In this regard, we pre-train an agent on a set of tasks from the Meta-World benchmark suite and adapt it to tasks from Continual-World. We conduct a comprehensive comparison of fine-tuning methods originating from supervised learning in our setup. Our findings show that fine-tuning is feasible, but for existing methods, performance on previously learned tasks often deteriorates. Therefore, we propose a novel approach, Learning-to-Modulate (L2M), that avoids forgetting by modulating the information flow of the pre-trained model. Our method outperforms existing fine-tuning approaches, and achieves state-of-the-art performance on the Continual-World benchmark. To facilitate future research in this direction, we collect datasets for all Meta-World tasks and make them publicly available.

## 1 INTRODUCTION

Reinforcement Learning (RL) has been instrumental in training agents capable of achieving notable successes, both in simulation, and in the real-world (Silver et al., 2016; Vinyals et al., 2019; Berner et al., 2019; Arjona-Medina et al., 2019; Bellemare et al., 2020; Degrave et al., 2022). However, such agents are usually highly specialized and incapable of performing well outside of a narrowly-defined task. Furthermore, adapting a pre-trained agent for a new task by fine-tuning usually results in decreased performance on prior tasks. This effect is well-known in the literature as *catastrophic forgetting* (McCloskey & Cohen, 1989).

A common paradigm to learn multiple tasks concurrently is multi-task learning (Caruana, 1997). However, typically, not all tasks we want an agent to learn are available at training time. In this case, new tasks must be learned in a sequential manner. Learning a new task ideally leverages knowledge from previously learned tasks and does not adversely affect the performance on them. A promising direction to obtain such agents is leveraging the Transformer architecture (Vaswani et al., 2017). Due to its capability of disentangling tasks, induced by the multi-head attention mechanism, such models excel at learning multiple tasks concurrently from large offline datasets (Lee et al., 2022; Reed et al., 2022; Jiang et al., 2022; Brohan et al., 2022). We want to leverage this capability in RL, thereby obtaining agents that can quickly learn new skills and tasks. However, it remains an open question how new tasks and capabilities can be added to the repertoire of an agent, without adversely affecting performance on previously learned tasks.

A common practice in supervised learning is to adapt pre-trained models to multiple downstream tasks. Established approaches include parameter efficient fine-tuning (PEFT) (Houlsby et al., 2019;

---

*Contact: `schmied@ml.jku.at`

Liu et al., 2022), and prompt-based tuning (PBT) approaches (Lester et al., 2021; Li & Liang, 2021). Both PBT, and PEFT, incorporate a small set of new parameters to adapt a pre-trained model to new tasks at minimum cost. Thus, these methods intrinsically avoid catastrophic forgetting. However, it is unclear how well these methods can be adopted for training RL agents from offline datasets. Current approaches to learn a wide range of tasks from offline datasets employ Behavioural Cloning (Bain & Sammut, 1995), or upside-down RL (Schmidhuber, 2019; Srivastava et al., 2019). These works include Multi-game Decision Transformers (MGDT, Lee et al., 2022), PromptDT (Xu et al., 2022), Gato (Reed et al., 2022) and VIMA (Jiang et al., 2022), which only consider full fine-tuning (FT) or PBT to adapt to new tasks.

Our goal is to identify methods that efficiently adapt to new tasks, while maintaining knowledge of prior tasks. We use the Meta-World/Continual-World (Yu et al., 2020; Wolczyk et al., 2021) benchmark and conduct a comprehensive evaluation of methods for adapting pre-trained Transformer models in an offline setting (Levine et al., 2020). We first pre-train a multi-task Decision Transformer (DT, Chen et al., 2021; Lee et al., 2022) before adapting the pre-trained model to new tasks using various FT, PEFT, and PBT methods. We find that FT methods adjust well to new tasks, but performance on previous tasks generally decreases. PBT, in contrast, retains performance on previous tasks but does not adapt well to new tasks. Therefore, we propose a new approach, Learning-to-Modulate (L2M), which is based on two recent methods from NLP and computer vision. Our method, L2M, learns a modulation pool that consists of keys associated with learnable modulation vectors. Therefore, we assume that tasks can be well discriminated, such that different tasks leverage different modulation vectors. Indeed, we observe such a clustering of tasks after

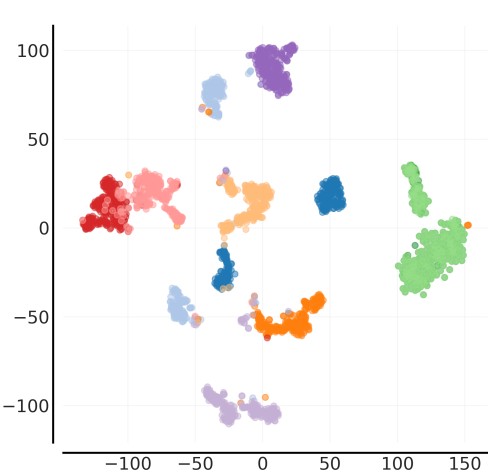

**Figure 1:** Visualization of t-SNE clustering of state embeddings for the first ten MT40 tasks. Similar tasks are clustered together, while dissimilar tasks are apart (legend is available in Appendix F).

pre-training in the learned embedding layers (Fig. 1) of our model. We show that L2M is capable of quickly adapting to new tasks, while maintaining performance on prior tasks. Moreover, L2M introduces only 0.21% additional parameters per task relative to the full model parameters.

To summarize, we make the following **contributions**:

- We conduct a broad evaluation of fine-tuning, parameter-efficient fine-tuning, and prompting methods for Transformers in RL.
- We propose L2M for efficient fine-tuning of a frozen pre-trained model by modulating the information flow via learnable modulation vectors. On Continual-World, our approach attains scores 27% higher than results in prior work, while merely adding 0.21% additional parameters per task.
- We release a dataset of trajectories collected in the Meta-World benchmark.

## 2 METHOD

Based on fine-tuning approaches from the field of NLP, we design a method tailored towards parameter-efficient task acquisition in Reinforcement Learning. Our aim is to avoid forgetting of previously acquired tasks.

### 2.1 BACKGROUND

We assume a Markov decision process (MDP) represented by the tuple $(\mathcal{S}, \mathcal{A}, \mathcal{P}, \mathcal{R})$. $\mathcal{S}$ and $\mathcal{A}$ denote state and action spaces, respectively, with states $s_t \in \mathcal{S}$ and actions $a_t \in \mathcal{A}$ at timestep $t$. The

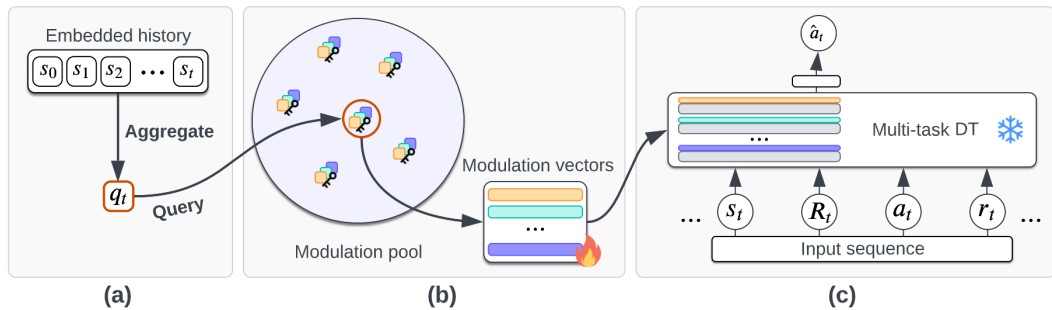

**Figure 2:** Illustration of L2M. **(a)** We construct a query $q_t$ by aggregating the embedded history of state-tokens. **(b)** The query $q_t$ is matched with learnable keys in a modulation pool, which map to learnable modulation vectors. We retrieve the modulation vectors with the highest similarity between $q_t$ and every key in the modulation pool. **(c)** Retrieved modulation vectors modify the pre-trained and frozen multi-task DT.

transition function $\mathcal{P}(s_{t+1} \mid s_t, a_t)$ takes a state-action pair and yields a probability distribution over next states. The reward function $\mathcal{R}(r_t \mid s_t, a_t)$ maps a given state-action pair to a scalar reward $r_t$. In addition, we assume access to a dataset $\mathcal{D} = \{\tau_i\} \mid_{i=1}^{B}$, where $\tau_i = (s_0, a_0, r_0, \ldots, s_T, a_T, r_T)$ represents a trajectory consisting of state-action-reward triplets and $B$ defines the number of trajectories in $\mathcal{D}$. Following Chen et al. (2021) and Lee et al. (2022), we augment each trajectory $\tau_i$ with *returns-to-go* $\hat{R}_t = \sum_{t'=t}^{T} r_{t'}$. We aim to find a policy $\pi_\theta(a_t \mid s_{t-K:t}, \hat{R}_{t-K:t}, a_{t-K:t-1}, r_{t-K:t-1})$ with parameters $\theta$ and context length $K$ that maximizes the expected return.

## 2.2 TRAINING DATASET

We train all methods on the Meta-World/Continual-World (Yu et al., 2020; Wolczyk et al., 2021) benchmark. This benchmark consists of 50 challenging robotic tasks, which we split into a set of 40 pre-training tasks and 10 evaluation tasks. In particular, we use Meta-World v2, instead of v1 used by Wolczyk et al. (2021), but will refer to it as Meta-World throughout this work. Details on the tasks and differences between Meta-World v1 and Meta-World v2 are provided in Appendix A. Meta-World comprises diverse tasks for robotic manipulation, such as grasping, manipulating objects, opening/closing a window, pushing buttons, locking/unlocking a door, and throwing a basketball. The state representation is similar across tasks to allow for transfer between them. The benchmark is designed to evaluate the agent's generalization ability to unseen tasks.

As Transformer-based methods in RL struggle with training in an online-setting, we collect datasets by training task-specific agents on all Meta-World tasks. We generate a large dataset of 10K trajectories for each of the 50 tasks[1]. The task-specific agents are trained via the Soft Actor Critic (SAC) algorithm (Haarnoja et al., 2018). The datasets comprise the entire replay buffer of the task-specific agents. Consequently, each dataset contains mixed behaviours, ranging from random to expert, depending on the performance of the task-specific agent. We choose this collection scheme since prior work has illustrated the benefits of training agents on data that comprises mixed behaviour (Lee et al., 2022).

Each pre-training dataset contains 10K trajectories of length 200, which amounts to 2 million transitions per task, i.e., state-RTG-action-reward tuples. Over all MT40 tasks, we get 80 million transitions (=320 million tokens) in total. For CW10, we follow the same procedure and obtain a dataset consisting of 20 million transitions, which corresponds to 80 million tokens.

## 2.3 LEARNING-TO-MODULATE (L2M)

L2M combines ideas for parameter-efficient fine-tuning from the NLP literature and prompting techniques from the computer vision literature. The first component, Infused Adapter by Inhibiting and Amplifying Inner Activations ((IA)[3], Liu et al., 2022), allows for PEFT by interleaving a pre-trained model with modulation vectors that are tuned for each new task. The second component,

---

[1]Available at: `https://github.com/ml-jku/L2M`

namely Learning-to-prompt (L2P, Wang et al., 2022c), learns a pool of prompts, such that each new task maps to a different prompt that is prepended to the model input.

Similar to L2P, we define a *modulation pool* that contains a set of $M$ keys, $\boldsymbol{K}_{pool} = \{\boldsymbol{k}_i\} \mid_{i=1}^{M}$. Each $\boldsymbol{k}_i$ is associated with a set of modulation vectors $\{\boldsymbol{l}_{k,b}^i, \boldsymbol{l}_{v,b}^i, \boldsymbol{l}_{ff,b}^i\}$ as values, for each layer block $b$ of a DT with $B$ layer blocks, where $\boldsymbol{l}_k \in \mathbb{R}^{d_k}$, $\boldsymbol{l}_v \in \mathbb{R}^{d_v}$, and $\boldsymbol{l}_{ff} \in \mathbb{R}^{d_{ff}}$. $d_k$, $d_v$, and $d_{ff}$ correspond to the dimensions of the keys, queries, and feed-forward activations in the DT, respectively. Since we follow a GPT-2-like architecture, $d_k = d_v$ and $d_{ff} = 4 * d_k$. We interleave each Transformer layer with separate modulation vectors, resulting in $d_k + d_v + 4 * d_{ff}$ learnable parameters per layer. At time $t$, we compose all states in a trajectory $\tau$ into a matrix $\boldsymbol{S}_{\leq t}$ after they are processed via the embedding layer of the DT. Subsequently, we reduce the matrix to a query vector $\boldsymbol{q}_t \in \mathbb{R}^{d_q}$ by an aggregation function $g(\cdot)$:

$$\boldsymbol{q}_t = g(\boldsymbol{S}_{\leq t}) \tag{1}$$

For the aggregation function $g(\cdot)$, we use mean-pooling by default. Further, we retrieve a set of modulation vectors $\{\boldsymbol{l}_{k,b}^j, \boldsymbol{l}_{v,b}^j, \boldsymbol{l}_{ff,b}^j\} \mid_{b=1}^{B}$ by the maximum similarity between each $\boldsymbol{k} \in \boldsymbol{K}_{pool}$ in the modulation pool and the query $\boldsymbol{q}_t$ at timestep $t$:

$$j = \underset{\boldsymbol{k} \in \boldsymbol{K}_p}{\arg\max} \operatorname{sim}(\boldsymbol{q}_t, \boldsymbol{k}) n(\boldsymbol{k})^{-1} \tag{2}$$

In our case, $\operatorname{sim}(\cdot, \cdot)$ corresponds to the cosine similarity and $n(\boldsymbol{k})^{-1}$ represents the inverse selection count for key $\boldsymbol{k}$ up to the current task. This way, we discourage that queries for different tasks attend to the same key. Subsequently, we use $\{\boldsymbol{l}_{k,b}^j, \boldsymbol{l}_{v,b}^j, \boldsymbol{l}_{ff,b}^j\} \mid_{b=1}^{B}$ to modulate the attention mechanism in the DT, as proposed by Liu et al. (2022):

$$(\boldsymbol{l}_v^j \odot \boldsymbol{V})^\top \operatorname{softmax}\left(\beta(\boldsymbol{l}_k^j \odot \boldsymbol{K})\boldsymbol{Q}\right) \tag{3}$$

Here, $\odot$ corresponds to element-wise multiplication, and $\beta = \frac{1}{\sqrt{d_k}}$. Also, $\boldsymbol{Q}, \boldsymbol{K}, \boldsymbol{V}$ refer to queries, keys, and values in the self-attention, respectively. Further, $\boldsymbol{l}_{ff}^j$ modulates the activations of the position-wise feed-forward activations in DT. All modulation vectors are initialized to ones, and, thus, the activations remain unchanged at the start of training. All keys in $\boldsymbol{K}_{pool}$ are initialized uniformly between $[-1, 1]$.

L2M unifies the benefits of both, (IA)$^3$ and L2P in the RL setting. It assures high-performance and few additional learnable parameters, while it avoids forgetting on the pre-trained tasks. Moreover, it provides a simple task-matching mechanism and enables scalability to numerous tasks.

## 3 EXPERIMENTS

We use the Meta-World benchmark to conduct a broad evaluation of established fine-tuning techniques in RL. We split the collected datasets into training and test sets. Particularly, we follow Wolczyk et al. (2021) and split the datasets into 40 Meta-World tasks (MT40) and 10 Continual-World tasks (CW10). The average performance of SAC across all tasks for MT40 and CW10 is shown in Table 1. Overall, 81% of all MT40 and 100% of CW10 tasks can be solved by the expert agent. More details on the training procedure, including hyperparameters, success rate distributions, individual task scores, and learning curves, are available in Appendix D.

First, we pre-train a DT on all MT40 datasets simultaneously, we refer to this as multi-task DT (see Section 3.1). Next, we conduct a broad evaluation of different FT, PEFT and PBT methods to adapt the pre-trained model to each of the CW10 tasks. We do not compare against meta-RL algorithms, as Mandi et al. (2022) showed that FT approaches perform on-par or better on several tasks. Section 3.2 shows results for fine-tuning on all CW10 tasks individually. Finally, we show results for training all methods in a continual RL setup, in which tasks are introduced sequentially (Section 3.3). Our experiment setup is illustrated in Figure 9 in Appendix C.

We report IQM and 95% bootstrapped confidence intervals, as proposed by Agarwal et al. (2021) (across 3 seeds per method and 50K bootstrap samples). In line with prior work on pre-trained models in RL (Lee et al., 2022; Jiang et al., 2022), we train our methods offline via return-conditioned upside-down RL to predict the next actions using the MSE loss. During evaluation, we set the target return to the maximum observed return in the respective dataset. We also found that a constant proxy of 2000 for the target return, yields similar performance.

| Dataset | Success Rate | Mean Reward |
|---------|--------------|-------------|
| MT40 | 0.84 ± 0.34 | 1414.62 ± 439.39 |
| CW10 | 1.0 ± 0.0 | 1540.49 ± 184.43 |

**Table 1:** Performance measures for data collection with SAC across MT40 and CW10 tasks. Mean and standard deviation are shown.

## 3.1 PRE-TRAINING

We pre-train a multi-task DT on all 80 million transitions of the generated MT40 datasets. We use the same underlying GPT-2-like network architecture as Lee et al. (2022). Implementation details and hyperparameters are provided in Appendix E. The aim is to have a single agent that can perform all 40 tasks. We experiment with different model sizes, varying the number of Transformer layers, number of heads per layer and embedding dimension (Table 5 and Figure 12 in the Appendix). We observe a saturation in performance beyond a certain number of parameters. The best performing architecture uses 4 layers, 8 heads and embedding dimension 512. We show strong performance (measured in the success rate) on the MT40 pre-training tasks (81%), but poor zero-shot generalization on CW10 (10%) (see Table 5, Appendix E).

We analyse the learned representations of the model via t-SNE (Van der Maaten & Hinton, 2008). Figure 1 illustrates emerging clusters of learned state embeddings for 10 tasks in MT40. We observe decent task separation. Similar tasks are clustered together (e.g., *button-press-v2* and *button-press-wall-v2*), whereas dissimilar tasks are well separated (e.g., *assembly-v2* and *bin-picking-v2*). We repeat this analysis for other token types and a larger set of tasks in Figures 13 and 14 in the Appendix, respectively. We give a detailed explanation of the clustering procedure in Appendix F.

## 3.2 SINGLE-TASK EXPERIMENTS

We evaluate the performance of various FT, PEFT and PBT strategies on CW10 in a single task setup. Overall, we compare a total of 10 methods: (1) Full fine-tuning (FT), (2) FT of action head (FT-head), (3) FT of last Transformer layer and action head (FT-last+head), (4) Adapters (Houlsby et al., 2019), (5) (IA)[3] (Liu et al., 2022), (6) Prompt-tuning (Lester et al., 2021), (7) Prefix-tuning (Li & Liang, 2021), (8) P-tuning v2 (Liu et al., 2021b), (9) PromptDT (Xu et al., 2022), and (10) VIMA (Jiang et al., 2022). A detailed list of hyperparameters and training details for each method are provided in Appendix G. We fine/prompt-tune the pre-trained model for each task individually, and the performance is aggregated over all 10 tasks. A more detailed description of each method is available in Appendix B.

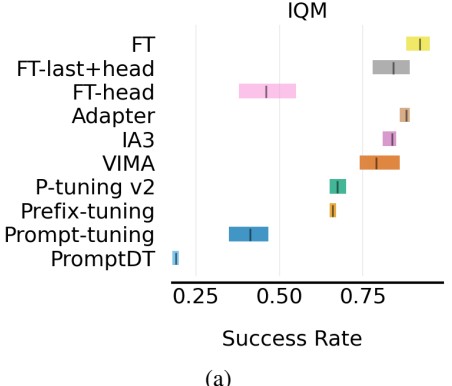
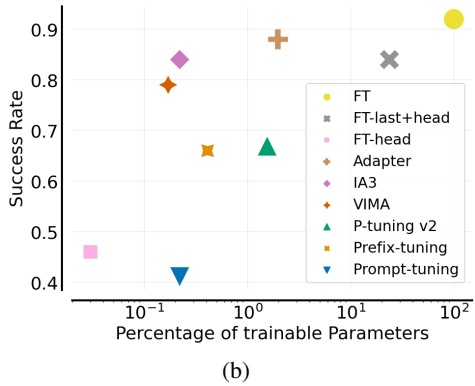

(a)  (b)

**Figure 3:** **(a)** IQM and 95% CIs of success rates for single-task experiments on CW10. Models have been pre-trained on MT40 and are then optimized for each CW10 task individually. We report the IQM over all 10 tasks in CW10. **(b)** Success rate vs. fraction of trained parameters trained for various fine-tuning techniques on single-task experiments on CW10.

Figure 3(a), shows the results for all 10 methods. Fine-tuning-based approaches attain considerably higher scores than prompt-tuning based approaches. This is expected, since FT utilizes the entire model capacity to learn the new task, while there is no guarantee for preserving knowledge about previously acquired tasks. Thereby, FT represents an upper bound in terms of performance. Adapters achieve the second-highest scores on average, followed by $(IA)^3$ and FT-last+head. Notably, regular prompt-tuning (i.e., prepending learnable prompts to the input) performs worse than merely fine-tuning the action head for each task.

We highlight the efficacy of the different methods by comparing the fraction of parameters trained against attained performance (see Figure 3(b)). For visualization purposes, we do not show PromptDT, since it does not introduce additional parameters and exhibits low performance. FT updates all parameters, while Adapters update two orders of magnitude less parameters (2%). Notably, $(IA)^3$, trains approximately the same amount of parameters as Prompt-tuning (0.21%), but attains drastically higher performance. This result indicates a trade-off between the number of trained parameters and performance. This trade-off is particularly apparent in a multi-task setup, as the number of additional parameters linearly scales with the number of new tasks to learn.

## 3.3 CONTINUAL-LEARNING EXPERIMENTS

Our goal is to adapt a pre-trained model to novel tasks, while mitigating forgetting of tasks learned during pre-training. However, in practice it is unlikely that training data for all novel tasks is available from the start but rather over time when new tasks are defined. Therefore, we adapt the pre-trained model to the CW10 tasks in a sequential manner, as proposed by Wolczyk et al. (2021). Moreover, we evaluate forgetting by measuring the performance on tasks acquired during pre-training after fine-tuning on all 10 tasks.

We compare the following methods: Full fine-tuning (FT), training task-specific action heads (FT-head), and tuning the last layer and task-specific action heads (FT-last+head). Additionally, we augment the PBT methods of the previous section with a prompt pool as in L2P, which enables learning of task-specific prompts. We refer to these methods as L2P + Prompt-tuning (L2P-PT), L2P + Prefix-tuning (L2P-PreT), L2P + P-tuning v2 (L2P-Pv2), and L2P + VIMA (L2P-VIMA). We note that L2P-PT reflects the original version of L2P proposed by Wang et al. (2022c). Additionally, we compare with two established methods from Continual RL, namely Elastic Weight Consolidation (EWC, Kirkpatrick et al., 2017), and L2 (Kirkpatrick et al., 2017). Moreover, we add another implementation of L2M, which is equipped with an oracle that provides information on what task is currently being observed. In line with Wolczyk et al. (2021), we use separate action heads per task for L2P and L2M. We do not show performance for Adapters and PromptDT. PromptDT learns prompts during pre-training, and thus, results in equal performance as for the

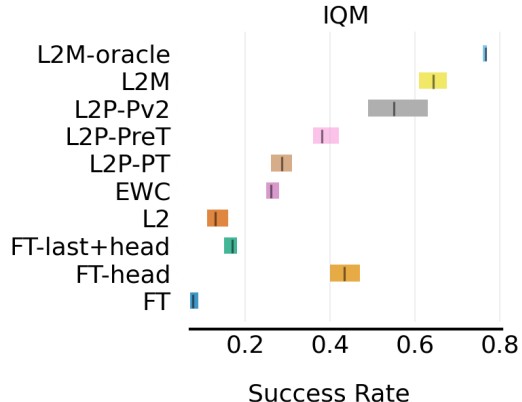

**Figure 4:** IQM and 95% CIs of success rates for CRL experiments on CW10. Models have been pre-trained on MT40 and are then trained on the tasks from CW10 sequentially. On each task, we train for 100K steps and then move to the next task in the sequence. We report the IQM over all 10 tasks in CW10.

single-task setup. Adapters perform well in the single-task setting, but incur much more parameters than L2M. For our 10 fine-tuning tasks, this amounts to approx. 20% of the parameters of the entire model. To illustrate an upper bound of performance, we again add two multi-task baselines, which train on all CW10 tasks simultaneously, either from scratch (FT-multi-task-scratch), or after the pre-training stage (FT-multi-task-pre-trained).

We train each method for 100K steps per task in CW10. After every 100K steps, we switch the dataset to the next in the task-sequence. We retain the same task sequence as Wolczyk et al. (2021) (see

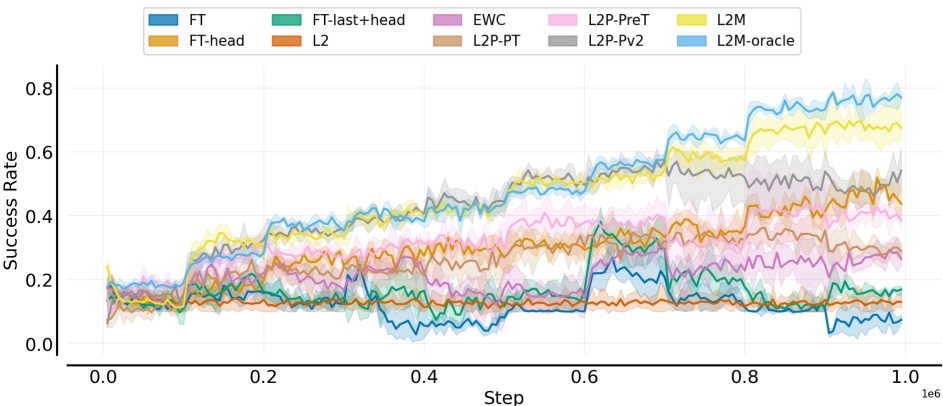

**Figure 5:** Learning curves for CRL experiments on CW10. On each task, we train for 100K steps and then move to the next task in the sequence.

Appendix A), and evaluate after every 5000 updates. Further training details and hyperparameters are shown in Appendix H.

In Figure 4, we show the success rates of all methods on CW10. In addition, in Figure 5 we present the corresponding learning curves during training. L2M outperforms all other approaches on CW10 and achieves an average success rate of 67% across all tasks. Adding a task oracle to L2M increases the success rate to 77% and approaches the single-task performance of (IA)[3] (84%). Similarly, as in the single-task setting, L2P combined with different prompting approaches performs significantly worse than L2M, except for L2P-Pv2 (55%). Interestingly, the established CRL method EWC performs poorly, merely reaching 27% average success. Further, L2 reaches only 13% average success, indicating that it learned a single task. Similar results for EWC were reported by Ben-Iwhiwhu et al. (2022). To the best of our knowledge, the results of our method L2M are the highest reported results on Continual-World v2 to date. Prior work reports average success rates of roughly 40% (Caccia et al., 2022; Ben-Iwhiwhu et al., 2022).

In addition, in Table 7 in Appendix H we report forgetting and rewards obtained at the end of training. As expected, FT and FT-last+head exhibit significant forgetting. Similarly, EWC suffers from a strong forgetting effect. L2 does not exhibit any forgetting, as it only learned to perform a single task in the first place. In contrast, FT-head reaches higher scores, indicating that the DT backbone learns features that generalize across tasks during pre-training. The PBT-based approaches perform well in terms of forgetting, however, achieve comparably low success rates. L2M achieves the highest reward, while mitigating forgetting. Overall, the pre-trained multi-task baseline achieves the highest success rate vs forgetting trade-off, which highlights the importance of the pre-training stage.

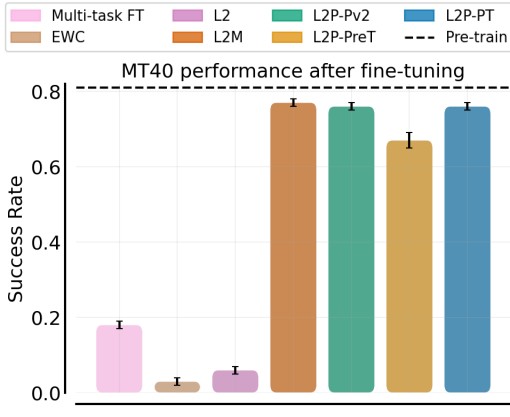

**Figure 6:** Performance on the MT40 pre-training tasks after fine-tuning on CW10.

Finally, we evaluate performance on the MT40 tasks after fine-tuning. To remain task-agnostic with L2M and L2P-based methods, we train a set of 40 keys for 50K steps on the same dataset as used for pre-training. During fine-tuning, we concatenate this set of keys to the remaining keys in the prompt pool. Figure 6 shows performance on the MT40 tasks. While multi-task FT achieves strong performance on CW10, the performance on MT40 experiences a severe drop to 18%. In contrast, L2M and L2P-based approaches maintain a similar performance level as prior to fine-tuning. Although we add extra keys for the pre-training tasks, the performance of L2M, and L2P approaches does not reach the performance after pre-training. We surmise this is due to conflation effects in the prompt pool and aim to investigate that in more detail in future work.

## 3.4  ABLATIONS

**Aggregation token.**  As specified in Section 2.3, we use embedded state-tokens aggregated over the sequence as query for the prompt pool in L2M and L2P. This design choice is inspired by the observed task separation in the embedding layer after pre-training (see Figure 1). We perform an ablation study on the choice of tokens representing the query (Figure 7). Indeed, we observe that using the state-token results in the best performance, as it aids task separation.  In contrast, using information of rewards or actions alone deteriorates performance.

**Limited data.** We repeat our main experiments in a limited data regime. We reduce the dataset size to 1%, 5%, 10% and 20% of the original size, which amounts to 20K transitions (= 100 trajectories), 100K (= 500 trajectories), 200K (= 1000 trajectories), and 400K (= 2000 trajectories), respectively. To account for the limited dataset size, we also reduce the number of update steps to 50K. In line with prior work in offline RL, the limited datasets are subsampled uniformly (Lee et al., 2022; Agarwal et al., 2020; Kumar et al., 2020).  Overall, we find that with fewer data points, performance degrades (see Appendix H). However, the overall performance ranking remains the same.

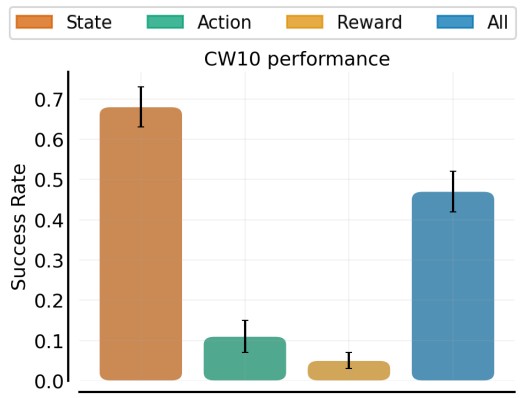

**Figure 7:** Ablation on the aggregation token used in L2M. Matching tasks based on state embeddings achieves the highest success rate.

**Excluding modulation vectors.** One important design choice in L2M is the placement of the modulation vectors $l_v$, $l_k$ and $l_{ff}$. Therefore, we conduct an ablation study in which we exclude either of the three modulation vectors when fine-tuning on CW10. In Figure 8, we present the results for all variants at the end of training. Removing $l_v$ or $l_k$ results in similar performance as regular L2M. In contrast, removing $l_{ff}$ results in worse performance (45.5%). Indeed, when dropping $l_v$ and $l_{ff}$ or $l_k$ and $l_{ff}$ in combination, we also observe a substantial decrease in performance. These findings suggest that modulating the attention mechanism is not as important as modulating the position-wise feed-forward layer.

**Pool size.** By default, we use a pool size (Section 2) of 20 and 50 for L2M and L2P, respectively. The size of the pool determines the potential overlap between task-specific prompts. Intuitively, a larger pool should result in less forgetting. To investigate the effect of larger pool sizes, we conduct ablation studies presented in Figures 19 and 20 in Appendix H. Indeed, we find that using a larger pool size results in better performance for both L2P and L2M. Contrary, reducing the pool size results in a severe drop in performance. L2M with pool size of 50 attains 76.7% average success rate, and thus comes close to multi-task FT. However, increasing the pool size also increases the number of additional parameters. Thus, the pool size trades-off performance and number of affordable additional parameters.

**Action head**. Using separate action heads makes the method reliant on task information. Ultimately, we aim for a CRL method that is task-agnostic. In turn, we conduct an ablation study with and without separate action heads per task for L2M. We report our results in Figure 21 in Appendix H.

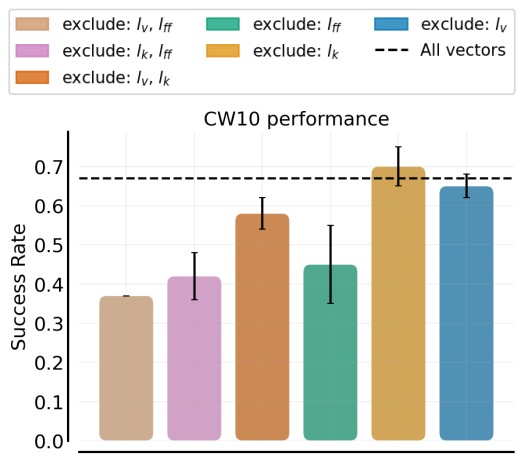

**Figure 8:** Performance on CW10 when excluding either of the three modulation vectors $l_v, l_{ff}, l_k$ (or combinations thereof).

Indeed, we do not find a significant performance difference between L2M with and without separate action heads. Thus, L2M is well suited for a task-agnostic setting.

## 4    RELATED WORK

**Transformers in RL.** Since the inception of Transformers (Vaswani et al., 2017) there has been a widespread adoption of the underlying architecture in many areas of Machine Learning research such as NLP (Devlin et al., 2019; Radford et al., 2019; Brown et al., 2020), computer vision (Dosovitskiy et al., 2021; He et al., 2022; Radford et al., 2021; Fürst et al., 2022; Ramesh et al., 2021; Rombach et al., 2022), speech recognition (Radford et al., 2022; Baevski et al., 2020) or video generation (Ho et al., 2022; Singer et al., 2022). More recently, Transformers have found their way into RL. Chen et al. (2021) introduced the Decision Transformer (DT), that obtained competitive results on the offline RL benchmark D4RL (Fu et al., 2020). Similarly, Trajectory Transformer (Janner et al., 2021) is based on GPT (Radford et al., 2018) and relies on dynamics modelling. Lee et al. (2022) extended DT to a multi-game setup and learn to play 46 Atari games. Meanwhile, a variety of DT-variants have been proposed (Zheng et al., 2022; Wang et al., 2022a; Shang et al., 2022; Meng et al., 2021). Siebenborn et al. (2022) replace the Transformer in DT with an LSTM (Hochreiter & Schmidhuber, 1997). PromptDT (Xu et al., 2022) demonstrated that prompting a pre-trained DT model with expert trajectories can improve the agent's ability to generalize to new tasks. However, the tasks they consider do not differ substantially (e.g., running forward vs. backward). Jiang et al. (2022) presented a prompt-based Transformer for robot manipulation, that integrates multi-modal prompts via cross-attention. Furthermore, Reed et al. (2022) trained a Transformer that scaled to over 600 tasks. Most recently, Brohan et al. (2022) presented a scalable Transformer for real-world robotics manipulation. Other works use a Transformer backbone for history compression in online RL (Parisotto et al., 2020; Paischer et al., 2022). Li et al. (2023) cover the landscape of Transformers in RL in more detail.

**Continual and multi-task RL.** Many CRL methods were proposed for computer vision, but can also be applied to a CRL setting. Early works include regularization approaches, such as EWC (Kirkpatrick et al., 2017), Synaptic Intelligence (Zenke et al., 2017), Memory-aware Synapses (Aljundi et al., 2018), Gradient Episodic Memory (GEM, Lopez-Paz & Ranzato, 2017) and A-GEM (Chaudhry et al., 2019). Other approaches rely on adding new modules to existing architecture (Rusu et al., 2016), iterative pruning (Mallya & Lazebnik, 2018), or improved exploration (Steinparz et al., 2022). A number of approaches have been tested on the Continual-World benchmark (Wolczyk et al., 2021), including 3RL (Caccia et al., 2022), ClonExSAC (Wolczyk et al., 2022) and Modulating masks (Ben-Iwhiwhu et al., 2022). More recently, Wang et al. (2022c) introduce L2P, which learns to prompt a frozen Vision Transformer (Dosovitskiy et al., 2021) and consistently outperforms prior methods on a number of CL benchmarks. Other follow-up works that rely on a prompting mechanism

for CL have been proposed (Wang et al., 2022b; Smith et al., 2022; Razdaibiedina et al., 2023). Recent works provide a comprehensive overview of the field of CRL (Hadsell et al., 2020; Lesort et al., 2020; Khetarpal et al., 2022; Baker et al., 2023).

**Parameter-efficient fine-tuning and Prompting.** Large-language models (LLMs) are pre-trained on vast amounts of data Devlin et al. (2019); Radford et al. (2019); Brown et al. (2020). After pre-training, it is desirable to specialize or fine-tune the foundation model (Bommasani et al., 2021) to a downstream task. A common way is fine-tuning all network weights or a fraction thereof (e.g., last layer or head). Fine-tuning the entire network is costly and/or may not be practical for multi-task settings. Parameter-efficient fine-tuning methods and prompt-based tuning offer attractive alternatives. Houlsby et al. (2019) repurposed Adapter modules (Rebuffi et al., 2017) to interleave pretrained Transformer layers. Variations thereof have been proposed (Bapna et al., 2019; Pfeiffer et al., 2021; 2020). Low Rank Adaptation injects trainable low-rank decomposition matrices into every layer of the model (Hu et al., 2022). Liu et al. (2022) proposed (IA)$^3$, which modulates the inner activation flow of the Transformer layers by elementwise multiplication with learned modulation vectors. Prompt tuning conditions a LLM by prepending learnable prompts to the embedded input sequence (Lester et al., 2021). Similarly, prefix-tuning adds learnable prefix vectors to the keys and values of each attention head input Li & Liang (2021). P-tuning v2, applies learnable prompts at each Transformer layer, not just at the input layer (Liu et al., 2021b). Liu et al. (2021a) give a comprehensive overview of prompt-based learning. Finally, a number of methods, such as Compacter (Karimi Mahabadi et al., 2021) and UniPELT (Mao et al., 2022), combine ideas from both categories.

## 5 CONCLUSION

Adapting agents to new tasks, while preserving performance on previously learned tasks, remains a major challenge towards more general RL agents. Consequently, we conduct a comprehensive evaluation of established fine-tuning methods for Transformers in RL. We evaluate both how well new tasks are learned and how strongly performance deteriorates on the pre-training tasks. While standard fine-tuning of a pre-trained model adapts well to new tasks, it suffers from severe forgetting. Prompt-based tuning techniques evade forgetting, while reaching lower performance on new tasks. We propose a novel method, L2M, which performs well in both dimensions. L2M efficiently fine-tunes the pre-trained model via learnable modulation vectors, and attains the best performance in terms of reward and forgetting. Also, L2M scales well for learning multiple new tasks and can be used in a task-agnostic manner.

Currently, our method uses a single set of modulation vectors for each task. In the future, we would like to investigate the compositionality of learned modulation vectors across tasks. We envision that associative memory methods, such as modern Hopfield networks (Ramsauer et al., 2021), which are able to store large sets of information (Widrich et al., 2020) and retrieve combinations thereof, will improve our method. We plan on incorporating such approaches in our method in future work. Also, while our current model is able to solve 50 challenging robotics tasks, the diversity of the training data is still modest. A number of additional offline datasets are available for RL (Fu et al., 2020; Gulcehre et al., 2020; Lu et al., 2022). We aim to expand our work to more domains, such as DMControl (Tassa et al., 2018), or Atari (Bellemare et al., 2013).

## ACKNOWLEDGMENTS

The ELLIS Unit Linz, the LIT AI Lab, the Institute for Machine Learning, are supported by the Federal State Upper Austria. IARAI is supported by Here Technologies. We thank the projects AI-MOTION (LIT-2018-6-YOU-212), AI-SNN (LIT-2018-6-YOU-214), DeepFlood (LIT-2019-8-YOU-213), Medical Cognitive Computing Center (MC3), INCONTROL-RL (FFG-881064), PRIMAL (FFG-873979), S3AI (FFG-872172), DL for GranularFlow (FFG-871302), EPILEPSIA (FFG-892171), AIRI FG 9-N (FWF-36284, FWF-36235), ELISE (H2020-ICT-2019-3 ID: 951847), Stars4Waters (HORIZON-CL6-2021-CLIMATE-01-01). We thank Audi.JKU Deep Learning Center, TGW LOGISTICS GROUP GMBH, University SAL Labs Initiative, FILL Gesellschaft mbH, Anyline GmbH, Google, ZF Friedrichshafen AG, Robert Bosch GmbH, UCB Biopharma SRL, Merck Healthcare KGaA, Verbund AG, GLS (Univ. Waterloo) Software Competence Center Hagenberg GmbH, TÜV Austria, Frauscher Sensonic and the NVIDIA Corporation.

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

## A  Environments

### A.1  Meta-World

For our experiments, we use environments from the Meta-World benchmark (Yu et al., 2020). Meta-World tasks consist of a Sawyer robotic arm simulated using the MuJoCo physics engine (Todorov et al., 2012). The set of all tasks in MT40 along with their descriptions are listed in Table 2.

The observation space is 39-dimensional. The first 18 dimensions the 3D position of the gripper, 1D normalized measure of gripper state, 3D position of the first object, 4D quaternion of the first object, 3D position of the second object, 4D quaternion of the second object. The next 18 values correspond to the same at the previous time step. The final 3 values correspond to the 3D position of the goal. The action space is 4-dimensional, with all actions in range $[-1, 1]$. The first three action dimensions correspond to the change in position of the gripper, and the final value represents the normalized force applied by the gripper (Yu et al., 2020). For each environment, the reward functions are manually engineered, and scaled to have a maximum value of 10 when the goal is achieved and a minimum of 0. The exact definitions are provided in Yu et al. (2020).

### A.2  Continual-World

The Continual-World benchmark was proposed by Wolczyk et al. (2021) and is built on top of Meta-World. Continual-World is a challenging CRL benchmark, and selects 10 of the 50 tasks contained in Meta-World. The task sequence used in Continual-World is known as CW10:

*hammer-v2*, *push-wall-v2*, *faucet-close-v2*, *push-back-v2*, *stick-pull-v2*, *stick-pull-v2*, *handle-press-side-v2*, *push-v2*, *shelf-place-v2*, *window-close-v2*, and *peg-unplug-side-v2*.

### A.3  Difference between Meta-World v1 and v2 environments

Continual-World benchmark was published using v1 of Meta-World. However, in the meantime v2 has been released. While the v1 environments had a 9 dimensional observation space, the observation space is 39-diemsional in v2. Another major change with the v2 environments is the introduction of dense reward functions, that guide the agent along the sequence of tasks: caging, gripping and moving in such a way that if at any point the agent fails, it will learn to try again. Further, the rewards are designed to lie in $[0, 10]$.

## B  Parameter-efficient fine-tuning and Prompting

**Fine-tuning.**  In full fine-tuning, the entire pre-trained model is trained on the new task. Other common variations thereof are fine-tuning the action head, the last layer or both together, which is also the choice in this work.

**Adapters.**  The parameter efficient use of adapters for transfer in the context of Transformers was proposed by Houlsby et al. (2019). The adapter consists of a down-projection, a non-linearity and an up-projection along with a skip connection. Two Adapter modules are added to every Transformer block. During training, only the Adapters (and optionally LayerNorms) are updated, while the rest is kept frozen. This significantly reduces the number of parameters to train, while preserving the ability

| Task | Description |
|------|-------------|
| assembly-v2 | Pick and place a nut onto a peg. |
| basketball-v2 | Pick and drop the ball into the basket. |
| bin-picking-v2 | Pick a puck from one bin and place into another. |
| box-close-v2 | Pick a lid and place on top of a box. |
| button-press-topdown-v2 | Press a button from the top. |
| button-press-topdown-wall-v2 | Press a button behind a wall from the top. |
| button-press-v2 | Press a button horizontally. |
| button-press-wall-v2 | Press a button behind a wall horizontally. |
| coffee-button-v2 | Push a button for coffee. |
| coffee-pull-v2 | Pull a mug from under the coffee machine. |
| coffee-push-v2 | Push a mug below the coffee machine. |
| dial-turn-v2 | Turn a dial by 180 degrees. |
| disassemble-v2 | Pick up a nut out of a peg. |
| door-close-v2 | Close a safe door. |
| door-lock-v2 | Lock the safe door by rotating the lock counter-clockwise. |
| door-open-v2 | Open a safe door. |
| door-unlock-v2 | Unlock the safe door by rotating the lock clockwise. |
| drawer-close-v2 | Close an open drawer. |
| drawer-open-v2 | Open a closed drawer. |
| faucet-open-v2 | Open a faucet by rotating it clockwise. |
| hand-insert-v2 | Insert the gripper into a hole. |
| handle-press-v2 | Press down on a handle. |
| handle-pull-side-v2 | Pull a handle up sideways. |
| handle-pull-v2 | Pull up a handle. |
| lever-pull-v2 | Pull up a lever by 90 degrees. |
| peg-insert-side-v2 | Insert a peg sideways into a hole. |
| pick-out-of-hole-v2 | Pick a puck out of a hole. |
| pick-place-v2 | Pick and place a puck at the goal spot. |
| pick-place-wall-v2 | Pick and place a puck at a goal spot behind a wall. |
| plate-slide-back-side-v2 | Slide a puck back, sideways out a goal. |
| plate-slide-back-v2 | Slide a puck back out of a goal. |
| plate-slide-side-v2 | Slide a puck sideways into a goal. |
| plate-slide-v2 | Slide a puck into a goal. |
| reach-v2 | Reach a goal position with the gripper. |
| reach-wall-v2 | Reach a goal position behind a wall with the gripper. |
| soccer-v2 | Push a ball into the goal. |
| stick-push-v2 | Pick up a stick and use it to push an object. |
| sweep-into-v2 | Sweep a puck into a hole. |
| sweep-v2 | Sweep a puck off the table. |
| window-open-v2 | Open a sliding window. |

**Table 2:** MT40 environment descriptions. Adapted from (Yu et al., 2020).

to adapt to new tasks. Houlsby et al. (2019) adapt a BERT (Devlin et al., 2019) model to 26 different text classification tasks, while training only roughly 4% of the parameters and attaining almost the same performance as full fine-tuning.

**(IA)$^3$: Infused Adapter by Inhibiting and Amplifying Inner Activations.** (IA)$^3$ was proposed by (Liu et al., 2022) and discussed in more detail in Section 2.

**Prompt-tuning.** Lester et al. (2021) learn soft prompts to enable a frozen pre-trained GPT-3 (Brown et al., 2020) model to generalize to downstream tasks. On tasks that involve domain shifts, prompt-tuning reduces overfitting and may even outperform full fine-tuning. In prompt-tuning, the learnable prompts are prepended to the input sequence, while the rest of the pre-trained model is kept frozen. This allows to effectively factor out the learnable task parameters from the model parameters.

**Prefix-tuning.** Prefix-tuning (Li & Liang, 2021) is another PBT-approach. Similar to prompt-tuning, prefix-tuning prepends learnable vectors (i.e., prompts) to the input. However, unlike in prompt-tuning, the learnable prompts are prepended to the keys and values of the attention head input. Therefore, prefix-tuning also incurs more parameters than prompt-tuning, but typically results in better performance.

**P-tuning v2.** Liu et al. (2021b) proposed P-tuning v2 as a successor of prefix-tuning. Instead of prepending continuous prompts only on the input layer, P-tuning v2 prepends them for every layer of the pre-trained model. This simple modification results in more parameters, but better performance. In particular, P-tuning v2 matches the performance of full fine-tuning on a number of NLP benchmarks and is effective across model sizes (330M to 10B parameters).

**PromptDT.** PromptDT (Xu et al., 2022) showcases few-shot adaptation capabilities of a pre-trained DT model on a number of MuJoCo tasks. In PromptDT the prompt is represented by an expert trajectory. During pre-training, the prompts are embedded via a learnable prompt embedding layer. As in regular prompt-tuning, the embedded prompts are prepended to the embedded input sequence. However, the considered tasks do not differ substantially (e.g., running forward vs. backward) and the PromptDT architecture requires changes to the pre-training stage.

**VIMA.** Jiang et al. (2022) presented VIMA, a prompt-based Transformer architecture for robot manipulation. In particular, VIMA integrates multi-modal prompts via cross-attention layers and operates on image-based inputs. The resulting architecture is capable of solving tasks like visual goal reaching and one-shot video imitation.

**EWC and L2.** EWC (Kirkpatrick et al., 2017) is an established regularization-based technique for continual learning. EWC helps to prevent forgetting of previous tasks when learning a new task by protecting parameters that are important for previously learned tasks. EWC uses the Fisher information matrix as a regularization term, which measures the sensitivity of each parameter with respect to each task and, thus, indicates which parameters need to be protected. L2 is used as a baseline in the original EWC publication (Kirkpatrick et al., 2017) and utilizes the L2 penalty to protect previously learned weights.

## C  EXPERIMENT SETUP

In Figure 9, we illustrate our experiment setup. We pre-train a multi-task DT on all MT40 datasets simultaneously (Section 3.1). Then we fine-tune the pre-trained model to 10 new tasks (CW10). We compare the fine-tuning performance in two setups. First, in a single-task setup in which we adapt the pre-trained model to each of the CW10 tasks individually (Section 3.2). Second, a continual RL setup, in which tasks are introduced sequentially to the pre-trained model (Section 3.3).

## D  DATA COLLECTION

To collect datasets, we select SAC as our expert agent of choice (Haarnoja et al., 2018). On each of the 50 tasks contained in Meta-World, we train a separate expert. We use the same network architecture as Wolczyk et al. (2021), 4 linear layers with 256 neurons, LayerNorm (Ba et al., 2016) after the first layer and LeakyReLU (Maas et al., 2013) activations, for both actor and critic. We use $\alpha = 0.01$ for the LeakyReLU instead of $\alpha = 0.2$ used by Wolczyk et al. (2021), as this choice performed better empirically. For each task, we train SAC for 2 million steps and record the entire

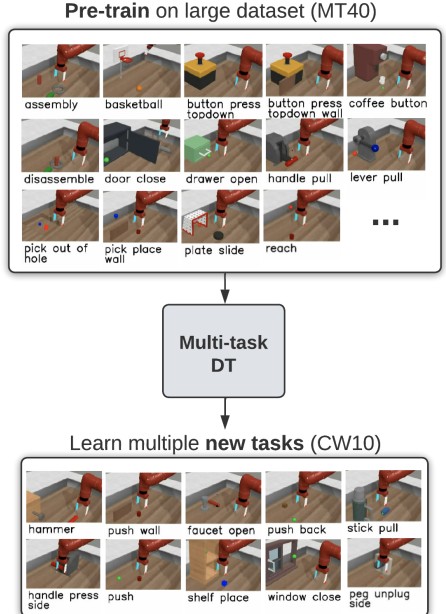

**Figure 9:** Illustration of our experiment setup. First, we pre-train a multi-task Decision Transformer on 40 Meta-World tasks (MT40). Then we fine-tune the pre-trained model to 10 new tasks (CW10). Environment images are adapted from Yu et al. (2020).

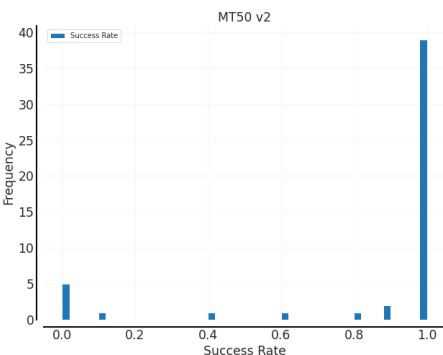

**Figure 10:** Histogram for success rates over all tasks. The majority of all MT50 tasks can be solved successfully at the end of training by single-task SAC, while on a minority of tasks the agents fails to achieve successful completion.

replay buffer. For every 50 interaction steps, we perform 50 gradient steps using Adam (Kingma & Ba, 2015)). We keep other parameters fixed at their default values in `stable-baselines3`. This includes the learning rate of $3e^{-4}$, batch size of 256, discount factor 0.99, and automatic entropy tuning. The target networks are synced after every update step with an update coefficient of 0.005. We evaluate the current policy after every 10K interaction steps. Each evaluation run consists of 10 individual evaluation episodes, and scores are average over all 10 episodes. The success rates already provide a measure of trajectory quality (TQ) for all individual datasets. Additional measures such as state-action coverage (SACo) can also be computed (Schweighofer et al., 2022).

The final datasets, contains 2 million transitions (i.e., state-RTG-action-reward tuples). This amounts to 100 million transitions in total for all 50 tasks (=400 million tokens), 80 million (=320 million tokens) and 20 million (=80 million tokens) for MT40 and CW10, respectively. Each trajectory is 200 steps long, and thus each dataset consists of 10K trajectories. In Tables 3 and 4, we list the success

| Task | Success Rate | Mean Reward |
|------|--------------|-------------|
| assembly-v2 | 0.0 | 1206.9 |
| basketball-v2 | 0.9 | 1375.95 |
| bin-picking-v2 | 0.0 | 474.81 |
| box-close-v2 | 0.0 | 759.15 |
| button-press-topdown-v2 | 1.0 | 1299.24 |
| button-press-topdown-wall-v2 | 1.0 | 1296.16 |
| button-press-v2 | 1.0 | 1430.44 |
| button-press-wall-v2 | 1.0 | 1508.16 |
| coffee-button-v2 | 1.0 | 1499.17 |
| coffee-pull-v2 | 1.0 | 1313.88 |
| coffee-push-v2 | 0.6 | 508.14 |
| dial-turn-v2 | 0.8 | 1674.29 |
| disassemble-v2 | 1.0 | 1396.55 |
| door-close-v2 | 1.0 | 1535.4 |
| door-lock-v2 | 1.0 | 1712.65 |
| door-open-v2 | 1.0 | 1544.32 |
| door-unlock-v2 | 1.0 | 1733.64 |
| drawer-close-v2 | 1.0 | 1845.92 |
| drawer-open-v2 | 1.0 | 1710.65 |
| faucet-open-v2 | 0.9 | 1727.98 |
| hand-insert-v2 | 1.0 | 1607.17 |
| handle-press-v2 | 1.0 | 1854.79 |
| handle-pull-side-v2 | 1.0 | 1613.72 |
| handle-pull-v2 | 1.0 | 1581.75 |
| lever-pull-v2 | 1.0 | 1449.05 |
| peg-insert-side-v2 | 1.0 | 1545.19 |
| pick-out-of-hole-v2 | 1.0 | 1435.64 |
| pick-place-v2 | 0.0 | 6.59 |
| pick-place-wall-v2 | 0.1 | 702.59 |
| plate-slide-back-side-v2 | 1.0 | 1766.24 |
| plate-slide-back-v2 | 1.0 | 1773.56 |
| plate-slide-side-v2 | 1.0 | 1663.35 |
| plate-slide-v2 | 1.0 | 1667.35 |
| reach-v2 | 1.0 | 1858.99 |
| reach-wall-v2 | 1.0 | 1831.14 |
| soccer-v2 | 0.4 | 445.84 |
| stick-push-v2 | 1.0 | 1470.71 |
| sweep-into-v2 | 1.0 | 1761.69 |
| sweep-v2 | 1.0 | 1458.35 |
| window-open-v2 | 1.0 | 1537.59 |
| Average | $0.84 \pm 0.34$ | $1414.62 \pm 439.39$ |

**Table 3:** Data collection on MT40.

rates and average rewards achieved on all MT40 and CW10 tasks, respectively. In addition, we show the success rate distribution and the learning curves on all 50 tasks in Figures 10 and 11, respectively. Across tasks, we observe notable differences in learning behaviour. The expert learns some tasks already after a few thousand interaction steps. On others, it takes much longer, if the expert succeeds at all. These differences in learning behaviour can be attributed to task difficulty. The harder the task, the longer it takes the agent to discover successful behaviour. For example, it is easier to open a door than to play basketball.

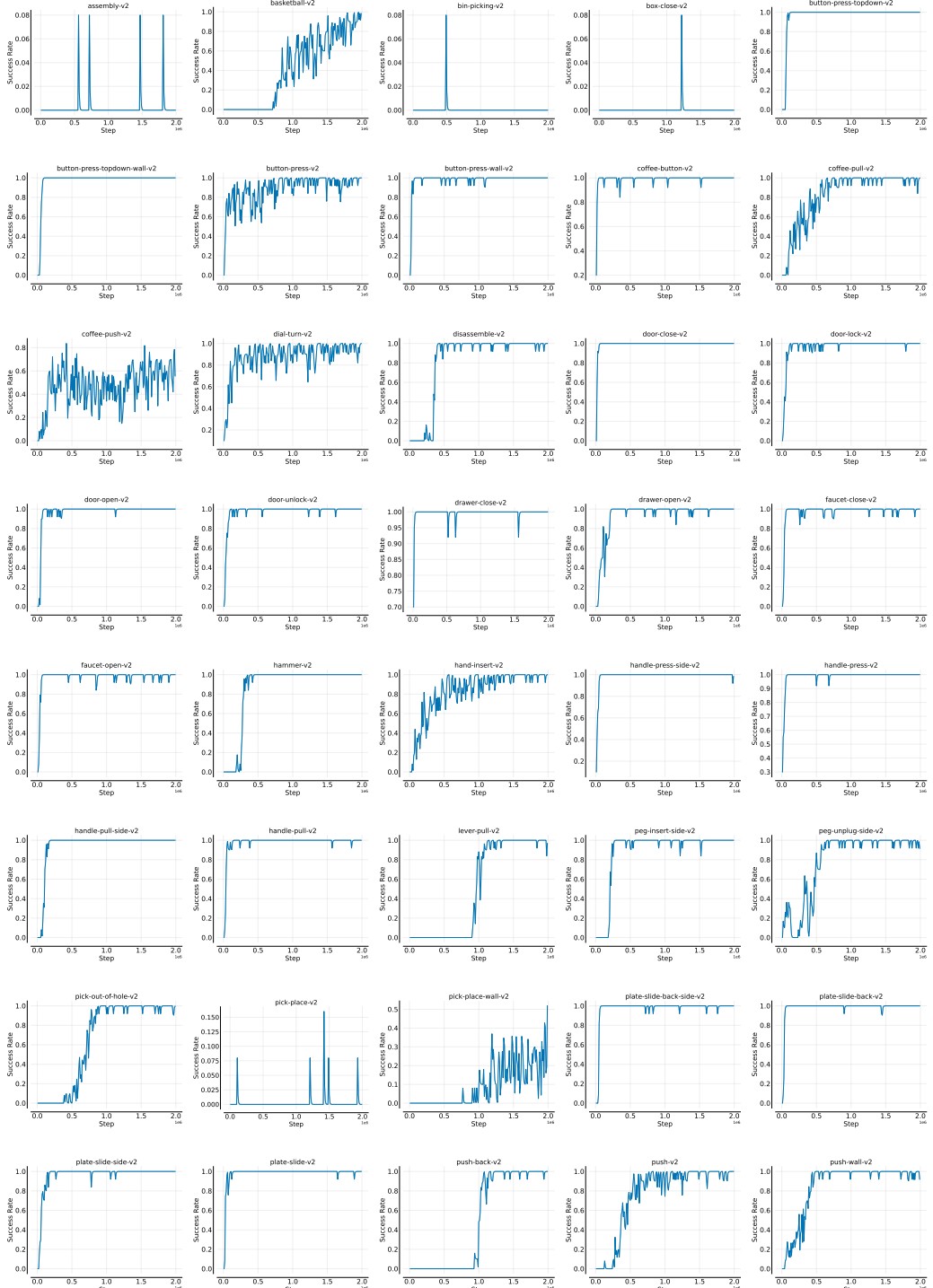

**Figure 11:** Learning curves for data-collection runs on all MT40 tasks with SAC.

| Task | Success Rate | Mean Reward |
|------|--------------|-------------|
| faucet-close-v2 | 1.0 | 1768.87 |
| hammer-v2 | 1.0 | 1632.21 |
| handle-press-side-v2 | 1.0 | 1842.17 |
| peg-unplug-side-v2 | 1.0 | 1373.45 |
| push-back-v2 | 1.0 | 1373.32 |
| push-v2 | 1.0 | 1672.88 |
| push-wall-v2 | 1.0 | 1594.37 |
| shelf-place-v2 | 1.0 | 1376.92 |
| stick-pull-v2 | 1.0 | 1344.29 |
| window-close-v2 | 1.0 | 1426.45 |
| Average | 1.0 ± 0.0 | 1540.49 ± 184.43 |

**Table 4:** Data collection on CW10.

| Layers | Heads | Embedding Dim | Reward | Parameters | Success Rate | Mean Reward |
|--------|-------|---------------|--------|------------|--------------|-------------|
| 1 | 2 | 128 | False | 368K | 0.62 | 1168.69 |
| 1 | 2 | 128 | True | 368K | 0.55 | 1063.69 |
| 3 | 2 | 256 | False | 2.7M | 0.81 | nan |
| 3 | 2 | 256 | True | 2.7M | 0.75 | nan |
| 3 | 4 | 256 | False | 2.7M | 0.78 | 1356.59 |
| 3 | 4 | 256 | True | 2.7M | 0.79 | 1359.49 |
| 4 | 4 | 512 | False | 13.3M | 0.75 | nan |
| 4 | 4 | 512 | True | 13.3M | 0.77 | nan |
| 4 | 8 | 512 | False | 13.3M | 0.8 | 1395.83 |
| 4 | 8 | 512 | True | 13.3M | 0.81 | 1398.04 |
| 6 | 12 | 768 | True | 43.5M | 0.8 | 1384.57 |

**Table 5:** Pre-training on MT40.

## E  PRE-TRAINING

We train our multi-task DT for a total of 1M update steps, with context length of 20 transitions (80 tokens). We use a learning rate of $3 \cdot 10^{-4}$, 1000 linear warm-up steps, gradient clip of 0.25, weight decay of 0.02, a batch size of 64 sequences, dropout of 0.2, and train using the AdamW optimizer (Loshchilov & Hutter, 2018). We base our implementation on the DT version in `Transformers`, and keep their default values for remaining parameters (Wolf et al., 2020) .

We use the same underlying GPT-2 like network architecture as Lee et al. (2022). We do, however, not make use of the proposed expert-action inference mechanism, as discussed in Section 3.1. Instead, we set the target return to the maximum return in the respective dataset and use a reward scale of 200 for all environments.

In Table 5 and Figure 12, we show the aggregate scores and learning curves across different model architectures. We vary the number of layers, the number of heads, the embedding dimension and whether reward conditioning is enabled. Overall, we do not observe considerable performance differences. Only for small models (365K) performance degrades.

## F  TASK SEPARATION ANALYSIS

To better understand what the DT learns, we visualize token embeddings using t-SNE (Van der Maaten & Hinton, 2008). First, we sample 256 state-RTG-action-reward sequences of length 20 (i.e., 80 tokens) for each task and record the activations of each (frozen) Transformer block. For each sequence, we then average the activations over the sequence. This can be done across the whole sequence or individually per token-type (i.e., for states, RTGs, actions, rewards). Each token has the same dimensionality. For the default multi-task DT model we use, the embedding dimension is set

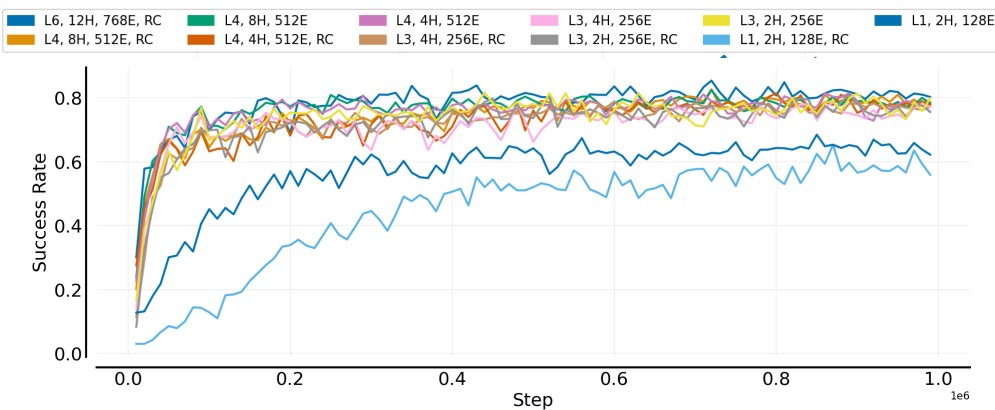

**Figure 12:** Learning curves for pre-training runs on MT40 with different architectural choices of the Transformer network (L = # of layers, H = # of heads, E = embedding dimension, RC = reward conditioned).

to 512. Thus, we end up with 256 512-dimensional vectors per task. We cluster all vectors using t-SNE. We use 2 embedding-space dimensions, perplexity of 30, 10K optimization steps, and the cosine distance.

In Figure 13, we show the t-SNE visualizations of all tokens combined, as well as for state, action, RTG, and reward tokens individually on the first 10 tasks in MT40. For states and all tokens, we observe good cluster separation. Also, for actions some clusters can be observed, but the overall separation is worse. In contrast, for rewards and RTG, we do not observe distinct clusters. This is expected, as we are only visualizing the embedding tokens. However, when moving up the layer hierarchy, we also observe decent cluster separation for reward and RTG tokens.

In Figure 14, we present the t-SNE visualization for state tokens on the first 20 tasks in MT40. Here we also observe good cluster separation.

## G    SINGLE-TASK EXPERIMENTS

We compare 10 different methods in this setting:

1. Full fine-tuning (FT)
2. FT of action head
3. FT of last Transformer layer and action head
4. Adapters (Houlsby et al., 2019)
5. $(IA)^3$ (Liu et al., 2022)
6. Prompt-tuning (Lester et al., 2021)
7. Prefix-tuning (Li & Liang, 2021)
8. P-tuning v2 (Liu et al., 2021b)
9. PromptDT (Xu et al., 2022)
10. VIMA (Jiang et al., 2022)

On each task, we train for 100K update steps. Every 5000 update steps, we evaluate the fine-tuned multi-task DT within the actual environment for 10 evaluation episodes and average performance across each evaluation episode. The final performance scores are aggregated over all CW10 tasks. We show the aggregated task performances in Table 6. In addition, we provide the respective learning curves in Figure 15. We test for statistical significance at the end of the training via a one-sided Wilcoxon rank-sum test (Wilcoxon, 1992) using a confidence level of $\alpha = 5\%$, and report the p-values in parentheses. FT obtains the highest score of 92% average success across all CW10 tasks

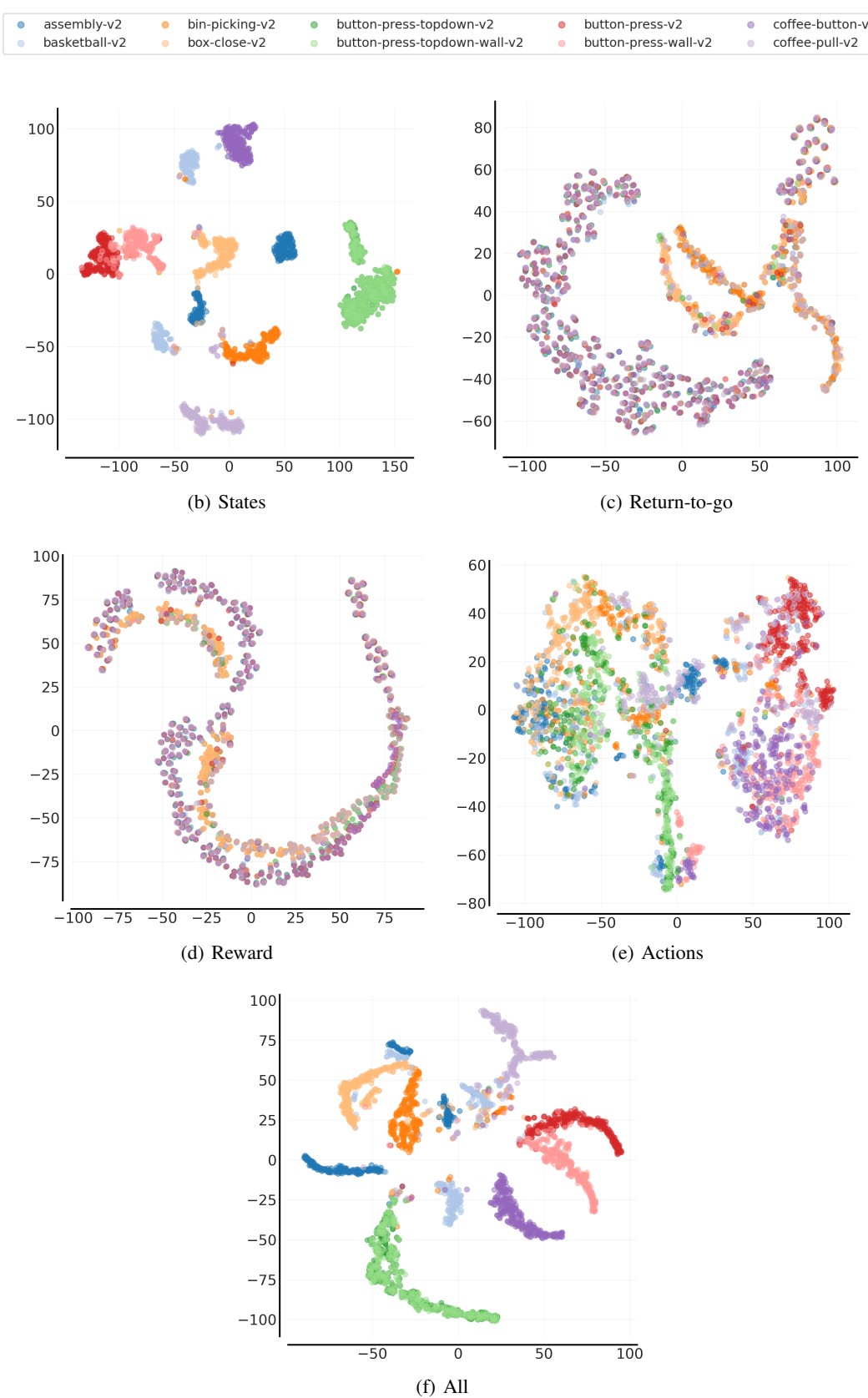

**Figure 13:** t-SNE visualization of state/RTG/action/reward embeddings in first Transformer block for the first ten tasks in MT40.

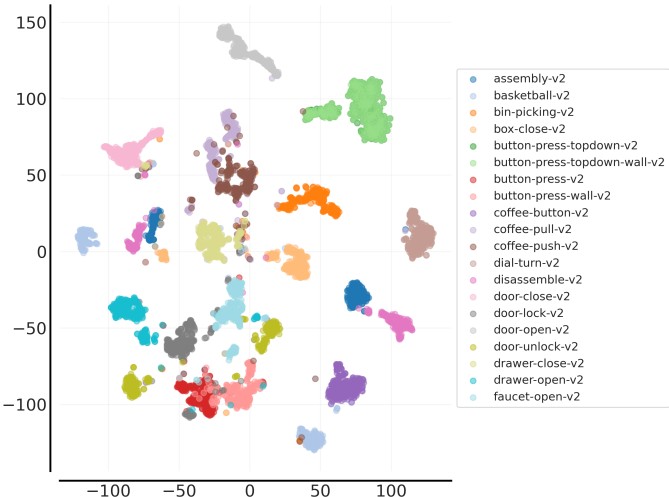

**Figure 14:** t-SNE visualization of state token embeddings in first Transformer block for the first twenty tasks in MT40.

| Method | Success Rate | Mean Reward |
|---|---|---|
| FT | $0.92 \pm 0.15$ | $1525.88 \pm 207.45$ |
| FT-last+head | $0.84 \pm 0.22$ | $1400.84 \pm 373.35$ |
| FT+head | $0.46 \pm 0.3$ | $860.87 \pm 495.58$ |
| Prefix-tuning | $0.66 \pm 0.31$ | $1158.22 \pm 434.45$ |
| P-tuning v2 | $0.67 \pm 0.35$ | $1208.78 \pm 485.07$ |
| Prompt-tuning | $0.41 \pm 0.43$ | $817.2 \pm 676.57$ |
| PromptDT | $0.19 \pm 0.01$ | $389.53 \pm 7.78$ |
| VIMA | $0.79 \pm 0.28$ | $1365.88 \pm 364.88$ |
| Adapters | $0.88 \pm 0.23$ | $1455.27 \pm 309.31$ |
| $(IA)^3$ | $0.84 \pm 0.28$ | $1404.17 \pm 362.49$ |
| FT multi-task-scratch | $0.85 \pm 0.07$ | $1481.39 \pm 66.03$ |
| FT multi-task-pre-trained | $0.95 \pm 0.02$ | $1560.89 \pm 19.71$ |

**Table 6:** Aggregate scores for all single-task experiments on the full dataset.

and significantly outperforms all other single-task methods, except for Adapters ($p = 0.49$), $(IA)^3$ ($p = 0.26$) and FT-last+head ($p = 0.11$).

The first 8 methods share the same pre-trained multi-task DT. PromptDT and VIMA require custom changes to the pre-training stage, and we train separate models of the same scale for each. At evaluation time, PromptDT is prompted with expert trajectories and, thus, does not require additional learning. VIMA, requires custom designed prompts at training and evaluation time. However, these do not exist in Meta-World (and notably most other benchmarks). In our version of VIMA, we use expert prompts at training time. For learning new tasks with VIMA, we employ prompt tuning for better performance, instead of expert prompts. In Table 6 we also provide the scores for two multi-task baselines, which train on all CW10 tasks at once. The first multi-task model trains on all CW10 tasks from scratch, the second multi-task model starts from the pre-trained MT40 model.

For FT variations and Adapters, we use a learning rate of $1e^{-4}$. For all other methods, we use a learning rate of $1e^{-3}$. For Adapters, we employ a reduction factor of 16 for the down-projection. For all prompt-tuning approaches, we use a prompt length of 25 and a dropout rate of 0.2. We found these values to be optimal in preliminary experiments. For Adapters and $(IA)^3$ we do not use Dropout (Srivastava et al., 2014).

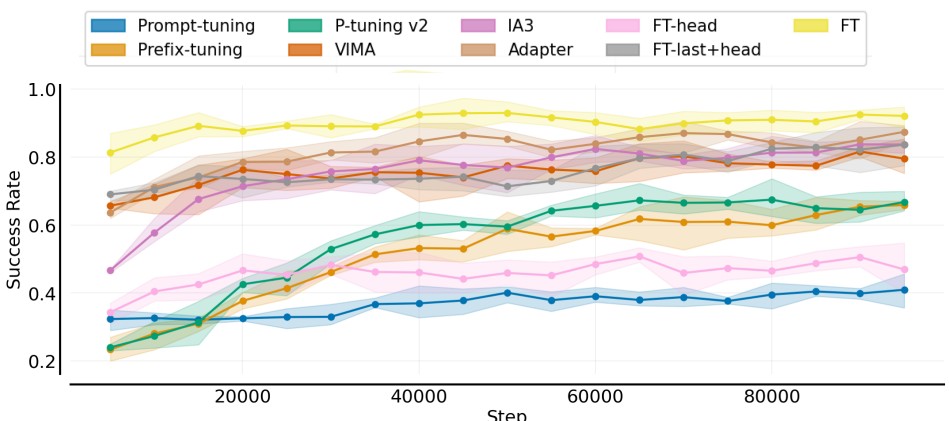

**Figure 15:** Learning curves for single-task experiments on the full dataset aggregated over all CW10 tasks.

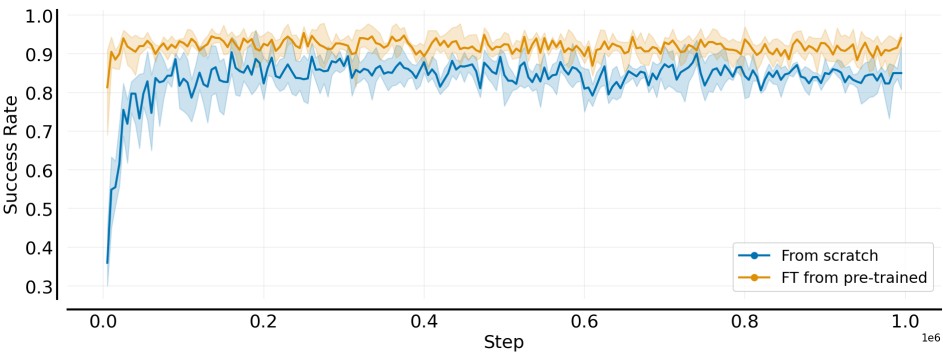

**Figure 16:** Learning curves for multi-task baselines on CW10, trained either from scratch or fine-tuned from the multi-task DT trained on MT40.

### G.1 MULTI-TASK TRAINING FROM SCRATCH VS. FROM PRE-TRAINED MODEL

In Table 6, we report the scores for model variants that train on all CW10 tasks at once in a multi-task fashion. While the first one is trained from scratch, the second one starts from the pre-trained model on MT40. In addition, we show the respective learning curves in Figure 16. The purpose of this experiment is to better understand whether pre-training on MT40 helps for fine-tuning on CW10. Indeed, we observe that the pre-trained model outperforms the learning curve of the model trained from scratch ($p = 0.092$), and attains 10% higher scores. Notably, the pre-trained model reaches its final performance only after a few thousand steps (first evaluation after 5000 steps).

## H CL EXPERIMENTS

### H.1 HYPERPARAMETERS

We compare 10 different methods in this setting:

1. Full FT,
2. FT of action head,
3. FT of last Transformer layer and action head,
4. L2P + Prompt-tuning,
5. L2P + Prefix-tuning,

| Method | Success Rate | Mean Reward | Forgetting |
|---|---|---|---|
| FT | 0.08 ± 0.01 | 219.32 ± 16.09 | 0.82 ± 0.33 |
| FT-last+head | 0.17 ± 0.02 | 308.57 ± 25.32 | 0.63 ± 0.3 |
| FT-head | 0.43 ± 0.04 | 832.3 ± 20.44 | - |
| EWC | 0.27 ± 0.02 | 525.31 ± 7.1 | 0.63 ± 0.35 |
| L2 | 0.13 ± 0.03 | 290.36 ± 50.17 | 0.0 ± 0.03 |
| L2P + prompt tuning | 0.29 ± 0.03 | 607.94 ± 68.51 | 0.14 ± 0.2 |
| L2P + prefix tuning | 0.38 ± 0.03 | 747.68 ± 46.05 | 0.14 ± 0.19 |
| L2P + P-tuning v2 | 0.55 ± 0.07 | 901.08 ± 116.05 | 0.17 ± 0.2 |
| L2P + VIMA | 0.25 ± 0.05 | 527.14 ± 92.92 | -0.07 ± 0.1 |
| L2M | 0.67 ± 0.03 | 1117.68 ± 41.06 | 0.04 ± 0.08 |
| L2M + oracle | 0.77 ± 0.01 | 1277.99 ± 23.23 | - |
| FT-multi-task-scratch | 0.85 ± 0.07 | 1481.39 ± 66.03 | - |
| FT-multi-task-pre-trained | 0.95 ± 0.02 | 1560.89 ± 19.71 | - |

**Table 7:** Continual RL experiments on full dataset.

6. L2P + P-tuning v2,

7. L2P + VIMA,

8. EWC (Kirkpatrick et al., 2017),

9. L2 (as used by Kirkpatrick et al. (2017)),

10. L2M.

We train each method for 100K steps per task in CW10, with a batch size of 64. After every 100K update steps, the policy switches to the next task in the sequence. We retain the same task sequence as Wolczyk et al. (2021) and do not reset the optimizer at task switches. Again, we evaluate after every 5000 update steps, but now evaluation is done on all 10 tasks. This results in 20K evaluation steps per evaluation run (10 evaluation runs per task × 10 tasks). For all L2P-based approaches, we use a prompt size of 25 and a prompt pool of 50 by default. For L2M, we use a pool size of 20. For EWC, we tune the value of the penalty (Figure 23).

## H.2 AGGREGATE RESULTS

We show the aggregate success rates, mean rewards obtained, and forgetting scores for all methods in Table 7. All metrics are reported at the end of training. The forgetting scores are computed as defined by Wolczyk et al. (2021). In Table 7, we also include the multi-task performance scores on all CW10 tasks (same as in Appendix G). Multi-task FT represents the upper bound in terms of performance. L2M achieves the highest success rates and significantly outperforms all other considered methods, except for L2P + P-tuning v2 ($p = 0.1$).

In Figure 17, we plot the performance against the percentage of parameters trained for each method. Full fine-tuning, EWC and L2 update 100% of all parameters. For FT and L2, we observe poor performance due to forgetting. EWC performs slightly better, yet suffers strongly from forgetting. Similarly, FT of the last layer and action head performs poorly, while updating around 24.2% of all parameters. FT of the action head only updates 0.61% of all parameters. Notably, L2M obtains the best overall results while only requiring 4.28% additional parameters for all 10 tasks. In contrast, L2P + P-tuning v2 already incurs 14.1%. However, we note that for all L2P and L2M-based approaches, the final percentage of parameters to store can be lowered. This is because not all parameters in the prompt pool may be used and, thus, unused ones can be discarded.

## H.3 AGGREGATION TOKEN ABLATION

For the main L2P and L2M results, we use embedded state-tokens aggregated over the entire sequence as input to the task-inference mechanism. This design choice is inspired by insights gained from the task separation analysis conducted in Section 3.1. Therefore, we conduct an ablation study on this design choice in Figure 18. We compare state-tokens to using action-token, reward-tokens, and

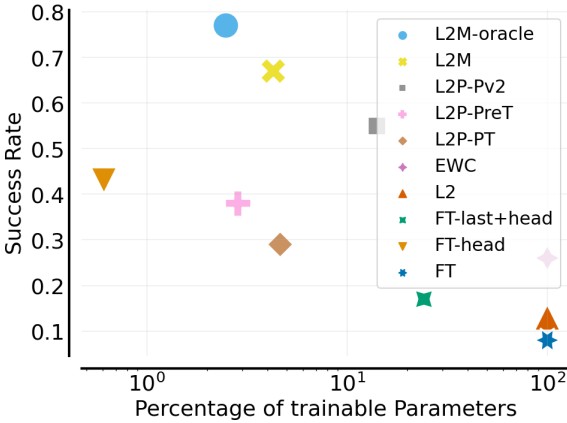

**Figure 17:** Performance vs. Percentage of parameters updated for CL experiments.

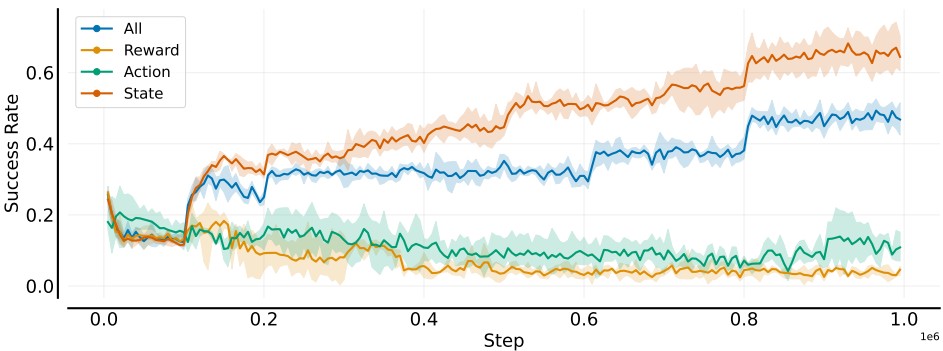

**Figure 18:** Aggregation token ablation.

all tokens combined. Indeed, we observe that using the state-token outperforms using all tokens ($p = 0.05$), action tokens ($p = 0.05$), and reward tokens ($p = 0.038$), as it aids task separation. In contrast, using information of rewards or actions alone does not suffice.

### H.4 POOL SIZE ABLATION

By default, we use a pool size of 20 and 50 for L2M and L2P, respectively. However, the size of the pool determines how much potential overlap there is between selected prompts or modulation vectors. A larger pool should result in less forgetting. To investigate this hypothesis, we conduct ablation studies presented in Figures 19 and 20. Indeed, we find that using a larger pool size results in better performance for both L2P and L2M. Conversely, a reduced pool size leads to decreased performance. L2M with pool size of 50 attains 76.7% average success rate, and thus comes close to the upper bound with task oracle. However, increasing the pool size also increases the number of additional parameters. Thus, the pool size trades-off performance and number of additional parameters one can afford.

### H.5 ACTION HEAD ABLATION IN L2M

All experiments with L2M and L2P, we present in this work, use separate action heads per task, following the standard setting by Wolczyk et al. (2021). Using separate action heads makes the method reliant on task information. Ideally, a CRL method should be task-agnostic, i.e., able to perform a task without being given prior information on what task has to be solved (He et al., 2019; Zeno et al., 2021; Caccia et al., 2022). To investigate this, we conduct an ablation study with and without separate action heads per task for L2M. We report our results in Figure 21. Indeed, we do

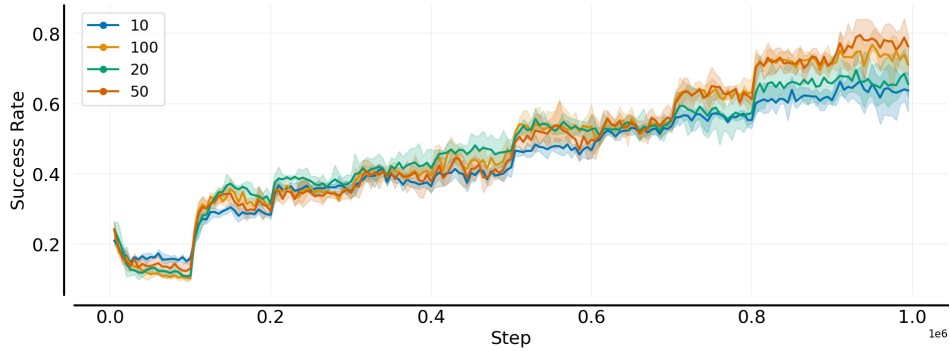

**Figure 19:** Pool size ablation for L2M. (Pool size=100 needs to be rerun, runs were cancelled.)

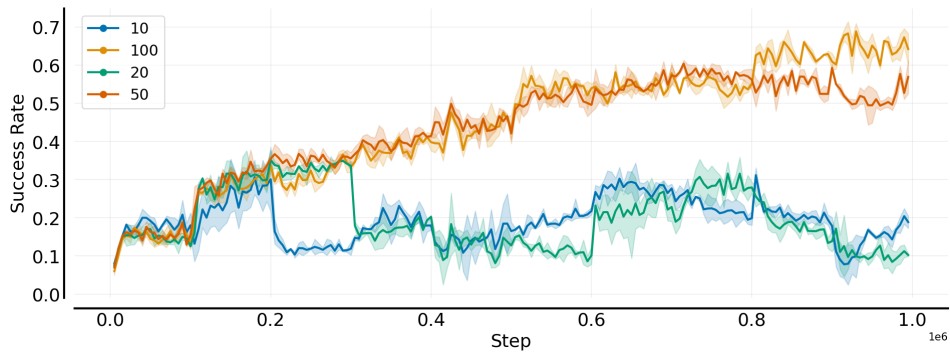

**Figure 20:** Pool size ablation for L2P + P-tuning v2.

not find a significant performance difference between L2M with and without separate action heads ($p = 0.35$). Thus, L2M is well suited for a task-agnostic setting.

### H.6 EXCLUDING MODULATION VECTORS IN L2M

We conduct another ablation study for L2M, in which we exclude either of the three modulation vectors $l_v$, $l_k$ and $l_{ff}$, for keys, values and hidden activation in the pointwise feedforward block, respectively. In Section, 3.3 and Figure 22, we already presented the main results of these experiments. In Figure 22, we additionally provide the learning curves for all variations. Overall, we observed that removing $l_v$ or $l_k$ results in similar performance as regular L2M across all CW10 tasks. We do not find a significant difference between excluding either of them ($p = 0.3$). In contrast, removing $l_{ff}$

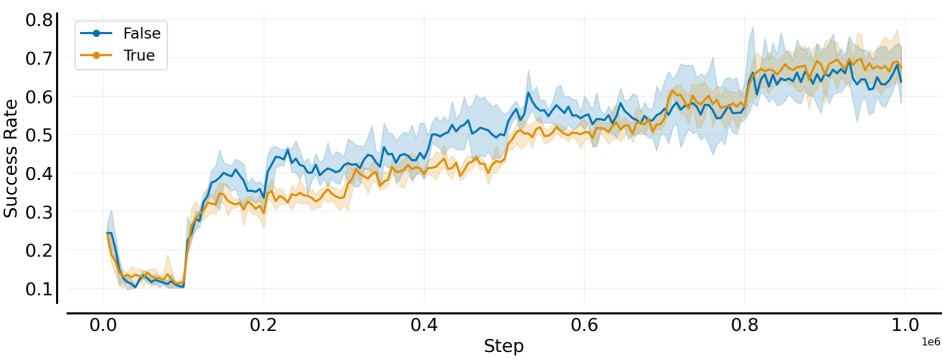

**Figure 21:** Head ablation.

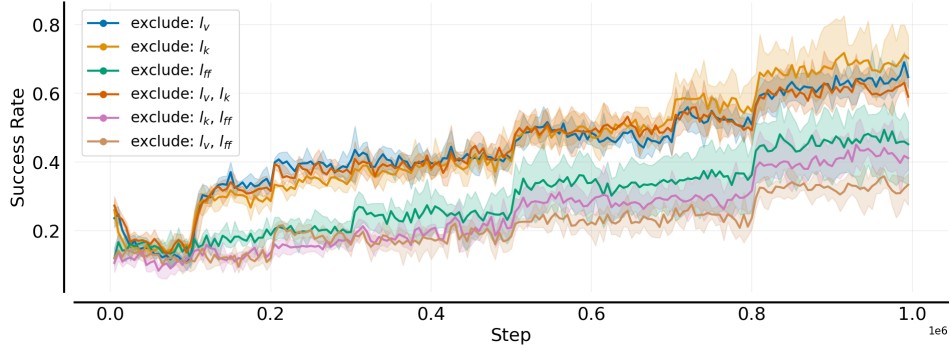

**Figure 22:** Excluding the modulation vectors $l_v$, $l_k$, or $l_{ff}$ in L2M.

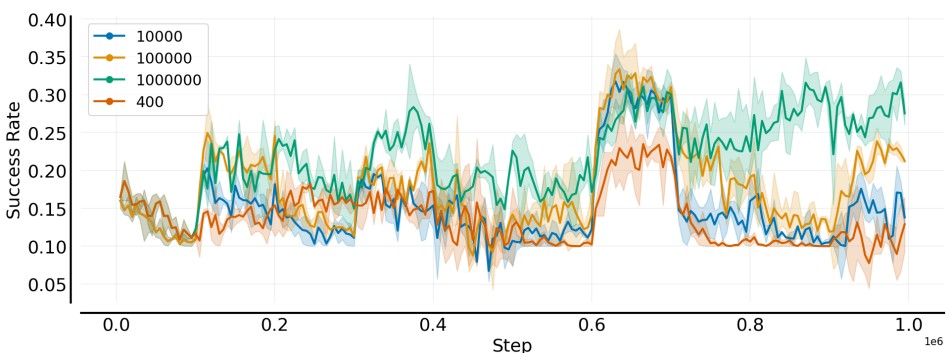

**Figure 23:** EWC ablation with different values for $\lambda$.

results in worse performance (45.5%). This indicates that $l_{ff}$ is the most important component in $(IA)^3$, at least in our experiments. Indeed, when dropping $l_v$ and $l_{ff}$ or $l_k$ and $l_{ff}$ in combination, we also observe a substantial performance decline. The opposite is true, when removing both $l_k$ and $l_v$ while preserving $l_{ff}$. This variant performs better, but worse, than regular L2M.

### H.7 LIMITED DATA ABLATION

In this ablation study, we analysed the effect of limited data on the performance of the compared methods. Therefore, we created four data subsets containing 1%, 5%,10% and 20% of the original dataset, respectively. We find that with fewer data points, performance degrades for all methods. Nevertheless, the relative performance ranking of the considered methods remains unchanged.

### H.8 EWC ABLATION

For EWC (Kirkpatrick et al., 2017), we compare different values for $\lambda = \{400, 1e^4, 1e^5, 1e^6\}$ . In Figure 23, we show the learning curves for all variations. When using a small value for $\lambda$, we observe strong forgetting, and, thus, poor overall performance. Only for $\lambda = 1e^6$ we observe signs of learning across tasks.

| Method | Dataset fraction | Success Rate | Mean Reward | Forgetting |
|---|---|---|---|---|
| FT-last+head | 1% | 0.04 ± 0.01 | 168.78 ± 24.25 | 0.54 ± 0.09 |
| FT-head | 1% | 0.5 ± 0.06 | 949.78 ± 65.18 | -0.03 ± 0.05 |
| L2P + prompt-tuning | 1% | 0.29 ± 0.05 | 635.94 ± 96.58 | 0.08 ± 0.15 |
| L2P + prefix-tuning | 1% | 0.35 ± 0.09 | 682.32 ± 154.63 | 0.11 ± 0.11 |
| L2P + P-tuning v2 | 1% | 0.43 ± 0.03 | 806.91 ± 103.69 | 0.14 ± 0.04 |
| L2M + oracle | 1% | 0.54 ± 0.06 | 992.36 ± 106.87 | 0.03 ± 0.09 |
| L2M | 1% | 0.49 ± 0.04 | 835.9 ± 100.89 | 0.0 ± 0.14 |
| FT-last+head | 5% | 0.12 ± 0.05 | 279.37 ± 73.9 | 0.65 ± 0.05 |
| FT-head | 5% | 0.49 ± 0.05 | 890.79 ± 52.18 | -0.08 ± 0.08 |
| L2P + prompt-tuning | 5% | 0.35 ± 0.09 | 629.52 ± 71.72 | 0.05 ± 0.09 |
| L2P + prefix-tuning | 5% | 0.36 ± 0.1 | 657.16 ± 128.83 | 0.1 ± 0.12 |
| L2P + P-tuning v2 | 5% | 0.43 ± 0.08 | 769.63 ± 123.33 | 0.17 ± 0.07 |
| L2M + oracle | 5% | 0.68 ± 0.08 | 1161.55 ± 119.87 | 0.02 ± 0.05 |
| L2M | 5% | 0.57 ± 0.09 | 966.07 ± 154.02 | 0.01 ± 0.03 |
| FT-last+head | 10% | 0.11 ± 0.04 | 249.32 ± 41.73 | 0.68 ± 0.05 |
| FT-head | 10% | 0.45 ± 0.06 | 896.46 ± 83.57 | 0.03 ± 0.03 |
| L2P + prompt-tuning | 10% | 0.32 ± 0.02 | 630.93 ± 76.26 | 0.04 ± 0.12 |
| L2P + prefix-tuning | 10% | 0.38 ± 0.09 | 703.31 ± 100.46 | 0.15 ± 0.04 |
| L2P + P-tuning v2 | 10% | 0.49 ± 0.06 | 818.6 ± 55.46 | 0.17 ± 0.07 |
| L2M + oracle | 10% | 0.72 ± 0.09 | 1202.89 ± 75.23 | -0.01 ± 0.03 |
| L2M | 10% | 0.55 ± 0.07 | 945.51 ± 162.3 | 0.05 ± 0.02 |
| FT | 20% | 0.07 ± 0.06 | 182.86 ± 29.48 | 0.79 ± 0.03 |
| FT-last+head | 20% | 0.19 ± 0.09 | 333.47 ± 58.62 | 0.65 ± 0.06 |
| FT-head | 20% | 0.47 ± 0.04 | 856.58 ± 87.19 | 0.02 ± 0.03 |
| L2p + prompt-tuning | 20% | 0.31 ± 0.06 | 618.33 ± 106.59 | 0.12 ± 0.07 |
| L2P + prefix-tuning | 20% | 0.39 ± 0.09 | 694.06 ± 96.73 | 0.15 ± 0.13 |
| L2P + P-tuning-v2 | 20% | 0.39 ± 0.06 | 762.86 ± 71.23 | 0.22 ± 0.1 |
| L2M + oracle | 20% | 0.73 ± 0.07 | 1214.22 ± 89.41 | -0.02 ± 0.06 |
| L2M | 20% | 0.66 ± 0.05 | 1066.36 ± 95.22 | 0.01 ± 0.05 |

**Table 8:** Continual RL experiments across dataset fractions. We compare 1%, 5%, 10%, 20%.

