# OpenReview forum: "Learning to Modulate pre-trained Models in RL"
_ICLR.cc/2023/Workshop/RRL — RRL 2023 Oral_

### Official Review · Reviewer_s1Nn · 2023-03-03

[review text omitted: it was posted to a different submission]

---

### Official Review · Program_Chairs · 2023-03-03
**Interesting paper, relevant to the workshop**

**Rating:** 4
**Confidence:** 4

**Review:**

Summary:
The authors focus on transferring pre-trained decision transformers from MetaWorld to ContinualWorld, in both one-step and continual reinforcement learning settings. They propose a hybrid prompting and modulation-based approach, combining prior works on this topic, which they show yields both good downstream performance and is robust to forgetting.

The paper provides a thorough set of baseline comparisons and ablations. The authors find that directly fine-tuning the model yields good single-task performance on downstream tasks, with performance clearly increasing as more parameters are fine-tuned. However, naive fine-tuning-based approaches lead to very poor performance when training sequentially on ContinualWorld tasks, apparently due to catastrophic forgetting. Fine-tuning only a restricted set of parameters yields both poor downstream performance and significant forgetting.

Relevance and significance: Very relevant to this workshop. The authors directly propose methods to reuse prior computation for expensively pre-trained models to solve new tasks. Good performance and thorough ablations indicate that this is a significant contribution.

Quality and clarity: Good. The paper is well-written and clear, with a very thorough appendix.

Additional feedback:
It might be good, just for the sake of completeness, to include a baseline where the agent is trained simultaneously on MetaWorld and ContinualWorld during fine-tuning. Since DT is inherently an offline algorithm this is fairly well-defined, and re-using buffers of previous tasks is a common technique in continual learning to mitigate catastrophic forgetting. This would be roughly equivalent to the FT-multi-task-pre-trained baseline but with MT-40 data also included.